# Surprise-minimization as a solution to the structural credit assignment problem

**Franz Wurm** [1,2,3] *, **Benjamin Ernst**[1], **Marco Steinhauser**[1]

**1** Catholic University of Eichstätt-Ingolstadt, Eichstätt, Germany, **2** Leiden University, Leiden, the Netherlands, **3** Leiden Institute for Brain and Cognition, Leiden University, Leiden, the Netherlands

* f.r.wurm@fsw.leidenuniv.nl

## Abstract

The structural credit assignment problem arises when the causal structure between actions and subsequent outcomes is hidden from direct observation. To solve this problem and enable goal-directed behavior, an agent has to infer structure and form a representation thereof. In the scope of this study, we investigate a possible solution in the human brain. We recorded behavioral and electrophysiological data from human participants in a novel variant of the bandit task, where multiple actions lead to multiple outcomes. Crucially, the mapping between actions and outcomes was hidden and not instructed to the participants. Human choice behavior revealed clear hallmarks of credit assignment and learning. Moreover, a computational model which formalizes action selection as the competition between multiple representations of the hidden structure was fit to account for participants data. Starting in a state of uncertainty about the correct representation, the central mechanism of this model is the arbitration of action control towards the representation which minimizes surprise about outcomes. Crucially, single-trial latent-variable analysis reveals that the neural patterns clearly support central quantitative predictions of this surprise minimization model. The results suggest that neural activity is not only related to reinforcement learning under correct as well as incorrect task representations but also reflects central mechanisms of credit assignment and behavioral arbitration.

## Author summary

In naturalistic environments, causal relationships between actions and their consequences are often hidden from direct observation. To overcome this structural credit-assignment problem, agents have to infer causal structures from experience. Here, we developed a computational model which formalizes action selection as the competition between structural representations, while action control is arbitrated towards the representation that minimizes surprise over time. To validate this model, we recorded behavioral and electrophysiological data from human participants in a novel task in which independent decisions are followed by outcomes, whereby the decision-outcome mapping is unknown. The model could account for patterns of choice behavior revealing clear hallmarks of credit assignment. Model-based analysis of EEG activity confirmed central model

---

**Data Availability Statement:** Custom MATLAB analysis scripts and de-identified human behavioral data are publicly available on Github (https://github.com/fwurm/202_Wurm_SurpriseMinimization.git). Preprocessed EEG data have been deposited at

---

Figshare and are publicly available (https://doi.org/
10.6084/m9.figshare.2055974.v1).

**Funding:** The author(s) received no specific
funding for this work.

**Competing interests:** The authors have declared
that no competing interests exist.

characteristics of concurrent prediction errors and a signature of evidence accumulation
and behavioral arbitration. These findings highlight a key role of surprise minimization
for both value and representation learning and reveal neural correlates of credit
assignment.

## Introduction

Reinforcement learning has become an indispensable framework for understanding reward-
based decision making and learning in the brain. Whereas computational models of reinforce-
ment learning provide a good mechanistic explanation for behavioral and neural data in both
animals and humans [1–3], they often make the simplified assumption that the structure
between causes (decisions) and consequences (outcomes) in the environment is fully known
and represented by the agent during learning. However, in complex, naturalistic environ-
ments, this causal structure is often hidden and not accessible to observation. To solve this so-
called structural credit assignment problem [4], an agent has to infer a representation that con-
tains information about which outcome has been caused by which decision, i.e., the correct
decision-outcome mapping. In the present study, we introduce a simple but efficient mecha-
nism which formalizes such inference about the hidden structure of the environment in terms
of a competition between multiple plausible representations of decision-outcome mappings
for action selection. Crucially, each representation is realized as an independent reinforcement
learning policy. Based on the idea that action selection should be controlled by the policy with
the highest prospect of reward [5–7], we propose that inference about the correct decision-out-
come representation can be derived from their estimated surprise signals which are used to
arbitrate control towards the policy minimizing surprise. We validate the model using behav-
ioral and neural data from a novel paradigm in which multiple independent decisions lead to
multiple outcomes while the correct decision-outcome mapping is unknown and must be
inferred.

Over the past few years, the issue of structural credit assignment got into the focus of rein-
forcement learning and decision making research and converging evidence highlights the
important contributions of the (intact) prefrontal cortex for solving it [8]. Various brain areas
such as the dorsolateral prefrontal cortex, the anterior cingulum and the orbitofrontal cortex
are implicated in the representation of the causal structure between decisions and outcomes
and the inference about hidden causes [9–15]. However, besides the importance of these dis-
tinct brain areas for solving the structural credit assignment problem in a fine-tuned network
structure, little is known about the distinct underlying computations. The emerging field of
representation learning is occupied with related problems, namely that of too much or too lit-
tle data for reinforcement learning [16]. If an agent is faced with too much data, selective atten-
tion could be used to reduce the dimensionality of the representation [17,18]. If an agents is
faced with too few data, causal inference could be used to generalize and distinguish between
environments [19,20]. While both mechanisms are important for explaining human represen-
tation learning, the present study will focus on the role of inference to reveal the structure in
an environment where the link between decisions and outcomes is hidden.

So far, inference has only played a subordinate role in the framework of reinforcement
learning. At its core, reinforcement learning formalizes how an agent optimizes its behavioral
policy with the goal to maximize reward and minimize punishment [21]. To deal with the
increasing demand for representations in computational models of reinforcement learning,
the existence of multiple parallel reinforcement learning policies in the human brain has been

postulated [3]. It is assumed that these policies compete for action selection and that the arbitration between them is instantiated by their respective reliability estimates [7]. Although this is a prominent principle in machine learning [6], evidence for such an arbitration mechanism within the human brain is sparse [22]. In this study, we build on the idea that reliability of a representation can be inferred from its respective prediction error. We introduce the notion of surprise as the absolute value of prediction errors within a policy [23] and propose that the comparison between surprise estimates can be interpreted as an evidence signal, so that the policy which minimizes surprise should receive an increasing weight for action selection.

To test the idea that representation learning and credit assignment in the human brain are based on surprise minimization and the arbitration between policies, we used a modified version of the bandit task (Fig 1A). On each trial, subjects in our multiple-bandits task had to make two independent decisions and were presented with two color-coded outcomes. Crucially, subjects were not instructed on the correct mapping between decision and outcome color. Therefore, to behave adaptively, the hidden structure of the task (i.e., the correct mapping between decision and outcome) had to be inferred and represented by repeatedly sampling from the environment. In Experiment 1, we ask whether and how participants learn to infer a correct representation of the task in this paradigm. By considering behavioral markers of credit assignment, we show that humans robustly infer the hidden structure of the multiple-bandits task and are able to explicitly act on its associated representation. As learning from representations is shown to be available to both implicit and explicit evaluations [24], we also confirm the emergence of explicit representations in a transfer task. By applying a computational model to these data, we show that surprise-based inference and the competition for control between policies can account for the behavioral pattern in our paradigm. In Experiment 2, we consider EEG data in our paradigm to extract neural correlates of two core mechanisms of our model: Prediction errors reflecting multiple concurrent decision-outcome representations, and both evidence signals and arbitration weights. The evidence signal is calculated on each trial from the surprise of these competing representations, whereas the arbitration weight is the accumulated evidence signal and scales the contribution of each representation to behavior.

## Results

### Human behavior shows hallmarks of implicit and explicit credit assignment

We first conducted a behavioral experiment (Experiment 1) to investigate whether participants show hallmarks of credit assignment in our paradigm, and to provide the empirical basis for our computational model. We recorded behavioral data of 48 human participants in the multiple-bandits task and a subsequent transfer task. In the multiple-bandits task, participants were asked to make two separate decisions in each trial (Fig 1C). Each decision required to choose between two distinct stimuli, which remained the same across a block of trials. After the second decision, two outcomes were presented that each consisted of values ranging between 1 and 99. Each outcome was determined by a separate Gaussian random walk so that within a decision, outcome values for the two possible actions always summed to 100 (anti-correlation; Fig 1B). Outcomes were color-coded (yellow and blue) and presented in parallel at a random, non-overlapping location on the display. Participants' task was to maximize outcomes by collecting points, as these were converted into real monetary reward at the end of the experiment. Crucially, whereas the general structure of the task was instructed, the correct mapping between decisions and outcome color was not instructed but had to be inferred. After each block, participants conducted a transfer task in which stimulus pairs were rearranged so that

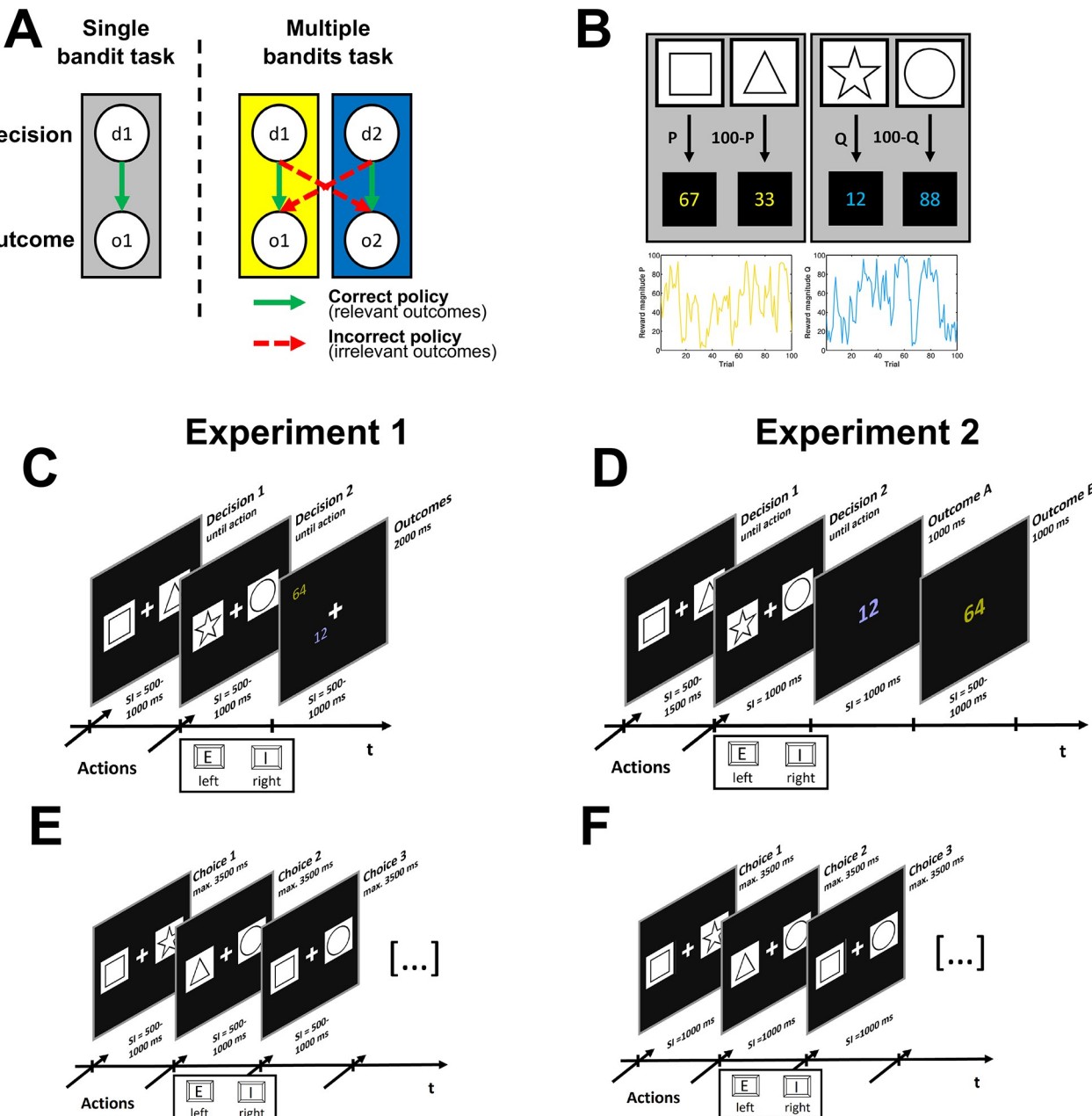

**Fig 1.** A. Schematic decision-outcome representations in two variants of the bandit task. In the single bandit task, one decision (d1) is followed by one outcome (o1). In the multiple-bandits task two decisions (d1, d2) are followed by two outcomes (o1, o2). White circles constitute the different states of the task. Gray and colored boxes indicate the true causal structure, called decision-outcome mapping. Colored arrows indicate the correct or incorrect policy, where correctness relates to the match between causal structure and an agent's representation. Outcomes are considered relevant if they belong to the correct representation and irrelevant if the belong to the incorrect representation. B. Graphical representation the multiple-bandits task, as implemented in the study. P and Q define the outcome value associated with each action. Over the course of a block, these values are subject to independent Gaussian random walks, as depicted in the boxes below. C and D. Trial representation of the multiple-bandits task in Experiment 1 and 2. E and F. Trial representation of the transfer task in Experiment 1 and 2. For the transfer task, stimuli from the different decision were mixed and participants were instructed to always choose the stimuli associated with a specific color (e.g., blue was associated with the star and the circle).

new stimulus pairs always contained one stimulus from each decision of the previous multiple-bandits task. Participants were asked to recall the correct decision-outcome mapping and to choose stimuli associated with a randomly determined feedback color. Performance in the

transfer task was taken as a measure of explicit credit assignment which complements the measure of implicit credit assignment from the multiple-bandits task.

We first analyzed choice behavior in the multiple-bandit task to investigate whether participants successfully inferred the correct mapping between decisions and outcomes.

Outcome-driven behavioral adaptation in a bandit task is characterized by a win-stay/lose-shift pattern. That is, agents tend to repeat their choice when the previous outcome for a given stimulus pair was a win but switch their choice if this outcome was a loss. We argued that win-stay/lose-shift behavior in the multiple-bandit task can be used as an index of implicit credit assignment in the following way. First, outcomes were defined as wins and losses if their value (ranging between 1 and 99) was larger or smaller than 50, respectively. As indicated by the Wilcoxon signed rank test, the resulting proportion of wins ($M = 0.64$, $SEM = 0.01$) significantly exceeded chance level, $Z = 5.91$, $p < .001$. Crucially, due to the nature of our modified bandit task with two decisions and two outcomes, each decision (stay or switch) can be classified separately for the correct policy and the incorrect policy (see Fig 1A). Hence, each decision on each trial is categorized according to the objectively correct policy (i.e., *relevant outcome win/loss*), and according to the objectively incorrect policy (i.e., *irrelevant outcome win/loss*). As the decision-outcome mapping is hidden from direct observation, we now consider how the probability of stay choices is influenced by the relevant and irrelevant outcomes as an implicit measure of credit assignment.

As shown by Fig 2A and a logistic regression analysis, the stay probability was higher when the previous outcome was a win than when it was a loss, and this held both for the relevant outcome ($z = 8.75$, $p < .001$), and for the irrelevant outcome ($z = 3.48$, $p < .001$). Relevant outcomes refer to outcomes that are based on the objectively correct decision-outcome mapping, whereas irrelevant outcomes link to decision only via the incorrect decision-outcome mapping. Thus, the results confirm our expectation that the relevant outcomes have a stronger impact on choice behavior than the irrelevant outcomes. Moreover, a significant interaction indicated that the effect of the irrelevant outcome was larger when the relevant outcome was a win ($z = 6.45$, $p < .001$), presumably reflecting that double wins led to a particularly strong tendency to repeat both decisions. While this analysis shows that decisions were influenced by both outcomes, successful credit assignment would require choice behavior to be more strongly driven by the relevant outcome than by the irrelevant outcome. To test this, we asked whether the relevant or irrelevant win outcome led to a higher stay probability by comparing stay probabilities after trials with a mixed outcome (relevant win/irrelevant loss vs. relevant loss/irrelevant win). A planned contrast revealed that choices were repeated more frequently when the relevant outcome alone signaled a win than when the irrelevant outcome alone signaled a win ($z = 5.23$, $p < .001$), a clear indicator that participants correctly assigned actions to their relevant outcomes.

Additionally, we investigated whether participants had acquired an explicit representation of the decision-outcome mapping at the end of each block by considering performance in the transfer task. With 68% ($SEM = 4\%$), the proportion of correct responses in the transfer task was significantly above chance level ($z = 4.01$, $p < .001$). This shows that participants formed an explicit representation of the decision-outcome mapping. Interestingly, the performance in the transfer task was highly correlated with both the average performance in the bandit task (proportion of wins; Pearson correlation, $r = 0.53$, $p < .001$; Fig 2C) and the strength of implicit credit assignment (relevant win/irrelevant loss minus relevant loss/irrelevant win; Pearson correlation, $r = 0.71$, $p < .001$; Fig 2E) across participants. Particularly the latter provides a first hint that the acquisition of an explicit decision-outcome mapping relies on the same mechanism that enables implicit credit assignment during task performance.

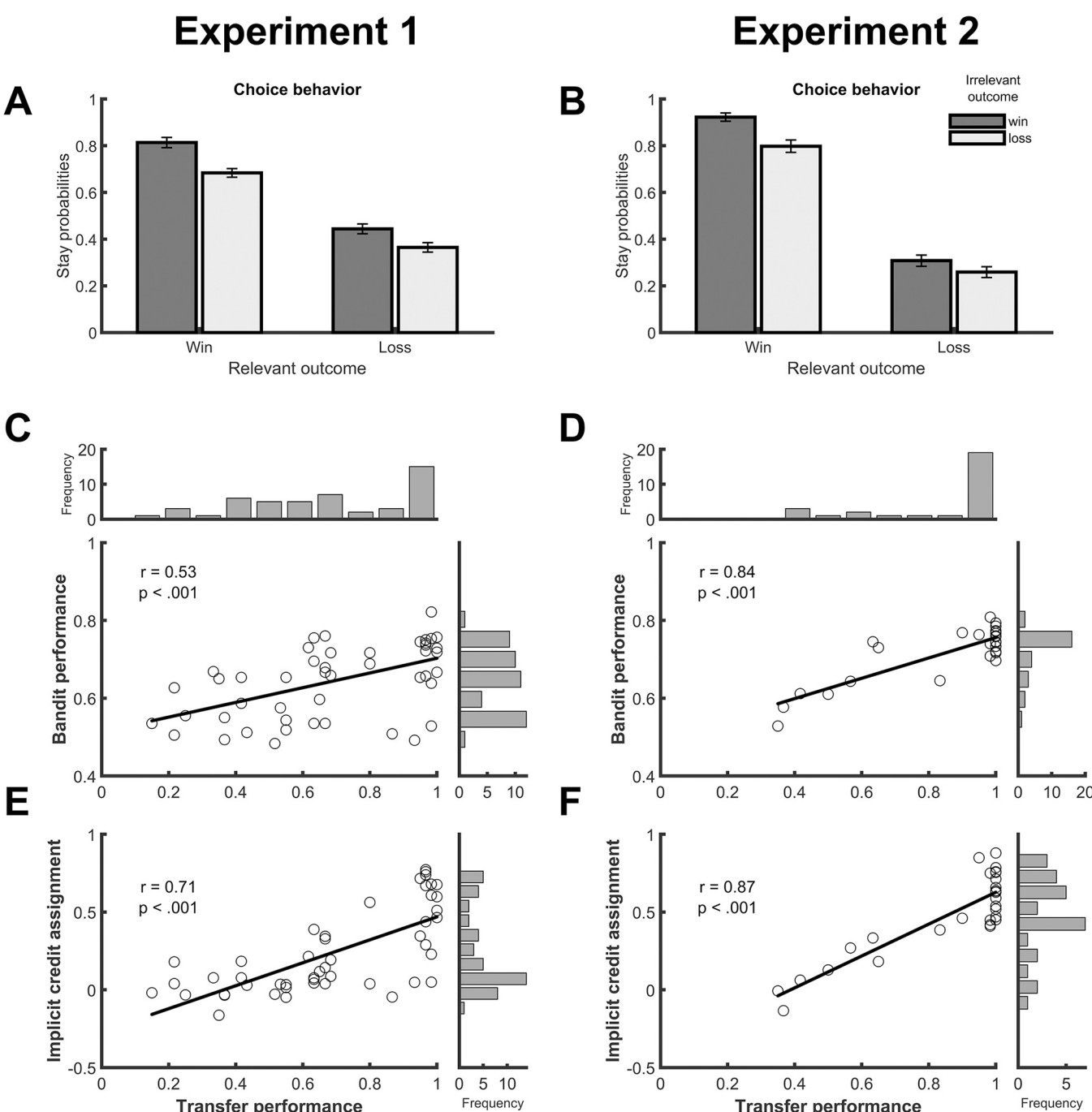

**Fig 2.** Behavioral results for Experiment 1 (left) and Experiment 2 (right). A,B. Stay probabilities following different combinations of relevant and irrelevant outcome. Relevant outcomes indicate the correct decision-outcome mapping, whereas irrelevant outcomes indicate the incorrect decision-outcome mapping. As the decision-outcome mapping is hidden from participants, stay probabilities for each decision are classified for both possible mappings (see Fig 1A). For example, decision 1 results in a win (blue), whereas decision 2 results in a loss (yellow). For decision 1, the relevant outcome (blue) is a win and the irrelevant outcome (yellow) a loss. Vice versa, for decision 2, the relevant outcome (yellow) is a loss and the irrelevant outcome (blue) is a win. C,D. Correlation between performance (probability correct) in the multiple-bandits task and the subsequent transfer task. Circles indicate participants. The black line indicates the regression line between both variables. E,F. Correlation between implicit credit assignment (stay probability of relevant win/irrelevant loss minus relevant loss/irrelevant win) in the multiple-bandits task and the subsequent transfer task.

## Surprise minimization enables successful credit assignment

In a next step, we constructed a computational model implementing the idea of surprise minimization and tested its ability to solve the structural credit assignment problem in our paradigm. The model follows a hierarchical architecture. At its core, multiple independent reinforcement learning policies compete for action selection with each policy corresponding to one of the two candidate decision-outcome mappings. Value updating within each policy follows the notation of temporal difference learning and is driven by prediction errors scaled according to a learning rate $\alpha$. Based on the anti-correlated nature of the task, unchosen options are also updated using the prediction error in a counterfactual manner. To arbitrate control towards the most plausible policy, the surprise minimization model contains a second-level inference mechanism which receives input from each policy in form of a surprise signal. Following earlier considerations [23,25], surprise is formalized as the absolute prediction error which provides a good estimate of the uncertainty about the expected outcome under each policy. The inference mechanism arbitrates control for action selection towards the policy with the smallest overall surprise. It does so by calculating an evidence signal that reflects the difference between surprise signals from the two policies. This evidence signal is then used to update the arbitration weight which determines the amount of control each policy is assigned during action selection. The impact of new evidence on the arbitration weight is controlled by the assignment rate $\varepsilon$. Action selection is then the weighted sum of each policy's value estimates normalized using a softmax procedure. The inverse temperature parameter $\beta$ controls for the stochasticity of this normalization process. In addition, the perseveration parameter $\rho$ reflects the stickiness of action selection independent of reinforcement learning. We used a Monte Carlo approach to demonstrate the effectiveness of the surprise minimization principle for credit assignment. For each of the following simulations, we conducted 1000 runs in the same instantiation of the multiple-bandits task as in the previous behavioral experiment with 3 blocks and 100 trials per each block. Parameters for each run ($\alpha, \varepsilon, \beta, \rho$) were randomly drawn from empirical prior distributions. A more detailed account of the simulations and the model is provided in the methods section.

Before exploring the full model, we first sought to validate the reinforcement learning processes that contribute to the hierarchical architecture at the first level. We simulated the model without the second-level inference mechanism and preset the arbitration weights either to the correct policy or to the incorrect policy. We hypothesized to find that only the relevant outcome influences the stay probability if the correct policy is applied whereas only the irrelevant outcome influences the stay probability if the incorrect policy is applied. Fig 3A and 3B confirms this expectation, thus demonstrating that our model successfully learns under each policy.

In a next simulation, we included the second-level inference mechanism. Rather similar to the empirical data, simulated patterns of win-stay/lose-shift behavior show a strong influence of the relevant outcome and a smaller effect of the irrelevant outcome (Fig 3C), thus demonstrating successful credit assignment. In line with our expectation, we identified differential surprise of the two first-level policies as the driving force for this shift towards the correct decision-outcome mapping. The distributions of surprise signals indicate that the correct policy was associated with smaller surprise values thus providing a more precise estimate of the true action values (Fig 3D, 1000 simulation, Wilcoxon signed rank test, $Z = 27.39$, $p < .001$). As a central computation of our model, this input is then converted into an evidence signal, which is instantiated as the difference between the arriving surprise signals. The accumulated evidence across trials is then used as a weight which arbitrates the control for action selection between first-level policies. An exemplary course of evidence is depicted in Fig 3E. Starting

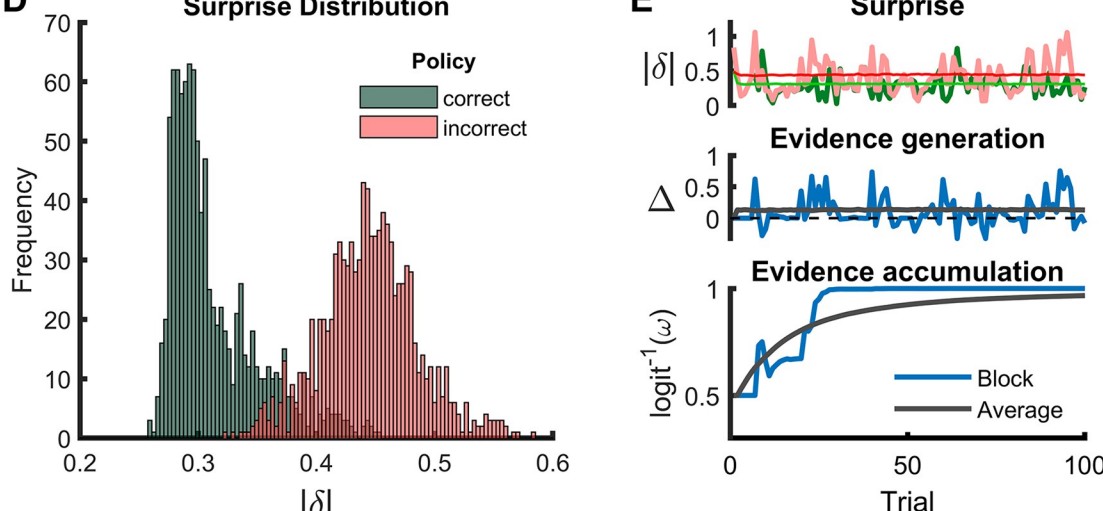

**Fig 3. Simulated data for the validation of the surprise minimization model.** A. Choice behavior when action selection is fully driven by the first-level policy representing the correct mapping between decisions and outcomes. B. Choice behavior when action selection is fully driven by the lower-level loop representing the incorrect mapping. C. Choice behavior when action selection is arbitrated between policies by the inference process of the surprise minimization model. D. Distribution of surprise signals calculated as the absolute prediction errors for both the correct and the incorrect policy. E. Illustration of the evidence

accumulation process. Surprise is calculated for both the correct (green) and incorrect (red) mapping (top panel). The evidence signal is calculated as the difference between these two surprise signals (middle panel). Accumulation of evidence and development of the arbitration weight (logit$^{-1}(\omega)$) over the course of a block (bottom panel). Starting in a state of uncertainty (0.5), the inference process gradually und robustly establishes the correct arbitration weight, leading to increasingly optimal credit assignment.

with no prior evidence about the correct decision-outcome mapping and the associated arbitration weight between policies (logit$^{-1}$($w_{corrmap}$) = logit$^{-1}$($w_{incorrmap}$) = 0.5), the continuous generation of evidence from the comparison of first-level surprise signals leads to gradual shift in weights. As surprise within a policy is influenced by random walk that drives the outcome, the process of evidence generation is noisy. However, the iterative accumulation of evidence leads to a successive and robust shift in arbitration weights towards the correct policy (logit$^{-1}$($w_{corrmap}$) ➜ 1). The robustness of this mechanism is also demonstrated by the fact that in 999 out of 1000 simulations, the final arbitration weight reflected a larger weight for the correct policy. Taken together, the simulation shows that the surprise minimization model leads to a reliable identification of the decision-outcome mapping in our multiple-bandits task, and thus successfully solves the structural credit assignment problem.

## Inference mechanism captures behavioral data both on an implicit and explicit level

After showing that surprise minimization can solve the credit assignment problem in the multiple-bandits task, we were interested in whether the model captures the empirical data better than four alternative models which perform the multiple-bandits task without an inference mechanism that supports credit assignment. The four alternative models share three parameters with the surprise minimization model (learning rate, inverse temperature, perseveration rate) but lack the fourth parameter, the assignment rate. Three alternative models were variants of the surprise minimization model in which the inference mechanism was switched off and the arbitration weights were set to a constant value. In the *correct* and *incorrect policy models*, control was arbitrated to the correct policy (logit$^{-1}$($w_{corrmap}$) = 1) or incorrect policy (logit$^{-1}$($w_{corrmap}$) = 0), respectively. In the *random policy model*, both policies were equally weighted (logit$^{-1}$($w_{corrmap}$) = 0.5) leaving the model in a lingering state of uncertainty. Finally, we considered a *joint action model* that integrates the two decisions into a single joint action. Instead of optimizing choice behavior for each decision separately, the model treats the two decisions as a single decision between four possible actions (i.e., all possible combinations between the two actions of each original decision), and action values are updated based on the sum of the two outcomes. We included this model because maximizing the total outcome could be a viable strategy in our paradigm when credit assignment is not possible. Please note, however, that such a strategy could not explain why participants acquire an explicit representation of the correct decision-outcome mapping as demonstrated in the transfer task.

The model selection statistics for the different candidate models are summarized in Table 1 (see Table A in S1 Supporting Information for additional model comparisons). In addition to standard information criteria (AIC, BIC), which provide a point estimate of a model's tradeoff between goodness of fit and complexity, we also included the exceedance probability, which is a quantification of the belief about the posterior probability [26]. As expected, model comparison points to the superiority of the surprise minimization model, as it outperforms the alternative models in all three measures despite its larger number of free parameters.

Additionally, we sought to elucidate the computational role of surprise in participants' decision-making. We hypothesized that the two central model variables—evidence and arbitration weight—play a crucial role in modulating the impact of expected action values of the different

**Table 1. Model comparison.**

| Model | # | Behavioral study | | | | EEG study | | | |
|---|---|---|---|---|---|---|---|---|---|
| | | -LL | BIC | AIC | xp | -LL | BIC | AIC | xp |
| Surprise–minimization | 4 | 14561 | 30218 | 29507 | 0.96 | 6666 | 13972 | 13557 | 0.99228 |
| Correct policy | 3 | 14969 | 30759 | 30226 | < .01 | 6858 | 14195 | 13884 | < .01 |
| Incorrect policy | 3 | 16650 | 34122 | 33589 | < .01 | 9133 | 18746 | 18435 | < .01 |
| Random policy | 3 | 15309 | 31440 | 30907 | 0.02 | 7726 | 15932 | 15621 | < .01 |
| Joint actions | 3 | 15497 | 31815 | 31281 | 0.03 | 79769 | 16432 | 16121 | < .01 |

Notes. # is the number of free parameters within a model, -LL is the negative log likelihood, BIC is the Bayesian Information Criterion, AIC is the Akaike Information Criterion, xp is the exceedance probability.

policies on future choice behavior. To test this assumption, we regressed participant's choice against the difference in action values (option A minus B), separately for the different policies ($V_{correct}$ and $V_{incorrect}$), ad-hoc evidence and accumulated arbitration weight. We found that future choices were predictable by value differences from both the correct policy ($z$ = -3.39, $p$ < .001) and the incorrect policy ($z$ = 4.88, $p$ < .001). Crucially, these main effects were qualified by significant interactions with arbitration weight. Therefore, we ran follow-up analysis, splitting the dataset into tertials [27] based on the arbitration weights. This revealed that the impact of action values on upcoming choice increased from low arbitration weights (estimate ±sem = 0.70, $z$ = 5.46, $p$ < .001) to high arbitration weights (1.93±0.19, $z$ = 10.08, $p$ < .001) for the correct policy. For the incorrect policy, we found the opposite pattern: the impact of action values decreased from low arbitration weights (0.77±0.10, $z$ = 7.83, $p$ < .001) to high arbitration weights (0.10±0.04, $z$ = 2.81, $p$ < .001). Evidence did not produce any significant effects on choice behavior (all $p$'s > 0.13). This shows clearly that accumulated evidence but not trial-to-trial evidence drives action selection and shifts action control towards the more plausible policy.

In a further effort to investigate the explanatory power of the surprise minimization model for explaining the behavioral data, the estimated model parameters were used to predict participants behavior in both the multiple-bandits task and transfer task. To this end, we calculated separate regressions in which the parameter estimates of the hierarchical inference model (learning rate $\alpha$, assignment rate $\varepsilon$, inverse temperature $\beta$, perseveration $\rho$) were used to predict the previously introduced behavioral measures. Fig 4A shows the resulting regression weights. For the implicit measure of credit assignment (stay probabilities relevant win/irrelevant loss minus relevant win/irrelevant loss), we obtained significant positive relationships for learning rate, $t(47)$ = 5.40, $p$ < .001, inverse temperature, $t(47)$ = 3.05, $p$ < .001, perseveration, $t(47)$ = -2.33, $p$ = 0.02, and assignment rate, $t(47)$ = 3.67, $p$ < .001. For the performance in the multiple-bandits task (proportion of wins), we observed significant positive relationships for learning rate, $t(47)$ = 8.99, $p$ < .001, inverse temperature, $t(47)$ = 6.38, $p$ < .001 and perseveration, $t(47)$ = -3.39, $p$ = 0.001. Finally, for the performance in the transfer task, we observed significant positive relationships between assignment rate, $t(47)$ = 2.71, $p$ = .009, as well as inverse temperature, $t(47)$ = 2.30, $p$ = .026, and assignment rate, $t(47)$ = -2.33, $p$ = 0.025. Thus, whereas task performance was driven by parameters related to the first-level policy, implicit and explicit credit assignment additionally depended on how strongly arbitration weights were adjusted based on evidence signals.

Taken together, these findings support the computational mechanism of surprise minimization and demonstrate that its implicit ability to assign credit generalizes to explicit metrics of credit assignment. This raises the possibility that explicit knowledge about the correct

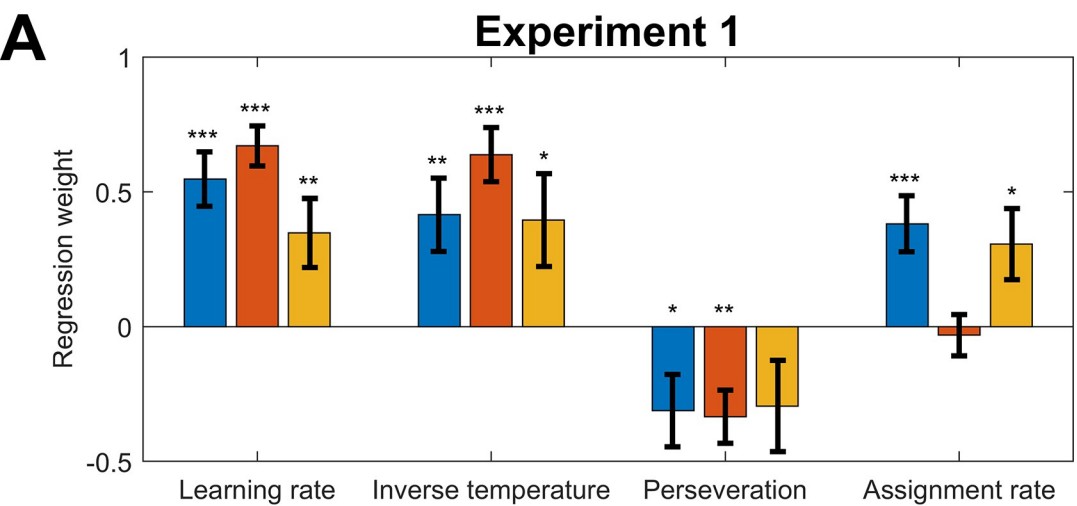

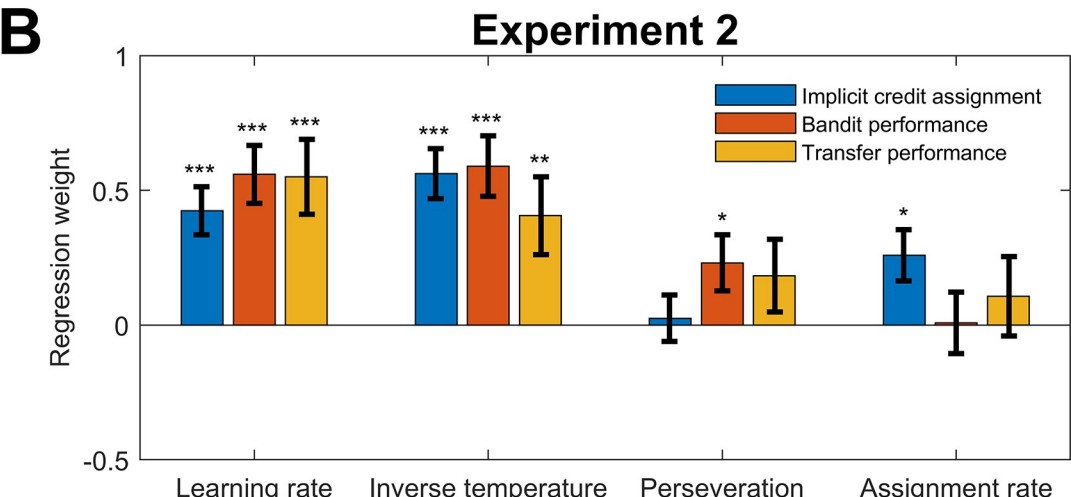

**Fig 4. Linear regression between the estimated parameters of the surprise minimization model (regressors) and the behavioral measures (criteria).** A. Regression values for Experiment 1. B. Regression values Experiment 2. Implicit credit assignment is the difference in stay probabilities between relevant win/irrelevant loss and relevant loss/irrelevant win.

decision-outcome mapping could have been generated by the same inference process that guided credit assignment in the bandit task. The task representation that minimized surprise and gained control over action selection could have been transformed into reportable knowledge about task structure. While the present data clearly show that such reportable knowledge has emerged, future studies could leverage similar experimental approaches to further reveal the mechanism that underlies our ability to translate experience into knowledge.

### Evidence for neural implementation of multiple concurrent first-level policies and evidence signals at feedback

The reported results provide support for our surprise minimization model by demonstrating that it can successfully account for implicit and explicit credit assignment in our task. However, the model also makes strong predictions regarding which computational mechanisms underlie credit assignment in the brain. First, it assumes that multiple policies related to

different decision-outcome mappings are concurrently evaluated. Second, it predicts that the brain calculates an evidence signal reflecting the difference between surprise signals associated with the involved policies. In a further experiment, we aimed to identify neural correlates of the computational mechanisms by considering neurophysiological responses during outcome processing in our paradigm. Due to its high temporal resolution, electroencephalography (EEG) seems to be a good method to achieve this, especially as it has repeatedly been shown to be sensitive to various forms of prediction errors [28–31]. In Experiment 2, we collected EEG data of 28 participants. The experiment utilized the same multiple-bandit task as in the previous experiment, the measurement of outcome-related brain activity required a modification of outcome displays. Whereas the color-coded outcomes in the behavioral study were presented simultaneously at non-overlapping locations, outcomes in the EEG study were presented one after another at the center of the screen. This procedure allowed to unambiguously assign neural activity to each presented outcome and also minimized the impact of eye movements and divided attention on multiple outcomes. To prevent that participants' decision-outcome assignment was biased by the order of outcomes, blue and yellow outcomes were presented in random order on each trial, while decisions were again presented in a fixed order.

Despite the slightly modified outcome displays (Fig 1D), we could replicate all results from Experiment 1. In the main task, the proportion of wins (M = 0.72, SEM = 0.01) again significantly exceeded chance level, rank test, Z = 4.62, p < .001. The logistic regression of stay probabilities (Fig 2B) showed significant effects of relevant outcome ($z = 9.85$, $p < .001$), irrelevant outcome ($z = 2.23$, $p = .026$), and the interaction between both ($z = 7.31$, $p < .001$). Again, stay probabilities were higher following relevant wins/irrelevant losses than following relevant losses/irrelevant wins using the Wilcoxon signed rank test ($z = 4.51$, $p < .001$), suggesting successful implicit credit assignment. Transfer task performance (M = 0.86, SEM = 0.04) was significantly above chance ($z = 4.43$, $p < .001$) and correlated with both bandit task performance ($r = 0.84$, $p < .001$; Fig 2D) and the strength of implicit credit assignment ($r = 0.87$, $p < .001$; Fig 2F). As in Experiment 1, model comparison points to the superiority of the surprise minimization model for explaining the behavioral data in Experiment 2, as it outperforms the alternative models in all three measures (see Table 1). We also predicted participant's choice behavior using the difference in action values (option A minus B), separately for the different policies ($V_{correct}$ and $V_{incorrect}$), ad-hoc evidence and accumulated arbitration weight. The results were in line with the findings from the behavioral experiment and show that the arbitration weights drive behavioral control towards the correct policy. For the correct policy, the impact of action values on upcoming choice increased from low arbitration weights (estimate ±sem = 1.04±0.08, $z = 11.81$, $p < .001$) to high arbitration weights (3.02±0.43, $z = 8.61$, $p < .001$). For the incorrect policy, the impact of action values on choice decreased from low arbitration weights (0.59±0.08, $z = 7.32$, $p < .001$) to high arbitration weights (0.16±0.06, $z = 2.85$, $p = .004$) for the incorrect policy. In a last step, we again used estimated model parameters to predict the behavioral measures of credit assignment (Fig 4B). For the implicit measure of credit assignment (stay probabilities relevant win/irrelevant loss minus relevant win/irrelevant loss), we obtained significant positive relationships for learning rate, t(27) = 4.75, p < .001, inverse temperature, t(27) = 6.05, p < .001, and assignment rate, t(27) = 2.73, p = .012. For the performance in the multiple-bandits task (proportion of wins), we observed significant positive relationships with learning rate, t(27) = 5.19, p < .001, inverse temperature, t(27) = 5.27, p < .001) and perseveration, t(27) = 2.21, p = .037. Finally, for the performance in the transfer task, we observed significant positive relationships with learning rate, t(27) = 3.96, p < .001, inverse temperature, t(27) = 2.81, p = .009 but not with assignment rate, t(27) = 0.73, p = .47.

We aimed to validate two central assumptions of our computational approach. First, the brain concurrently applies multiple policies related to the two possible decision-outcome

mappings. According to our model, we should find evidence that prediction errors of both the correct and the incorrect policy are reflected in brain activity after feedback. Besides the concurrent application of multiple policies, our computational model also predicts the existence of an evidence signal calculated as the difference between the surprise signals of the two policies. This evidence signal is of essential importance for credit assignment in the model, as it drives the trial-to-trial updates of the arbitration weight that determines the extent of control assigned to each first-level policy. To investigate the hypothesis that the central model variables are reflected in the human EEG signal, we utilized a model-based analysis of brain activity, i.e., we estimated single-trial values of the central model variables: prediction errors for the correct and incorrect policy, evidence signals and the arbitration weights ($\text{logit}^{-1}(\omega)$). We regressed these values on single-trial EEG data elicited for each trial and locked to the outcome. The resulting regression weights for each electrode, time step and participants were subjected to a permutation test to identify significant clusters of neural activity reflecting the latent variables (alpha level = 0.05; Fig 5A and 5B).

We found that reward prediction errors for both the correct and incorrect policy were reflected in the EEG data (Fig 5A, p's = .005), as revealed by the cluster-based permutation test procedure. For the correct policy, the cluster extended from approximately 492 to 773 ms over posterior electrode sites. For the incorrect policy, the cluster extended around 39 to 422 ms over parietal electrodes. The cluster-based permutation procedure also produced a significant effect for the evidence signal (Fig 5B, p < .001). The cluster extended from 273 to 797 ms and shifted from fronto-central to parieto-central locations. The arbitration weight regressor did not produce a significant cluster (p > .11).

In a more exploratory analysis, we investigated if the neural activity also reflects unsigned prediction errors. In line with a previous publication [32], the prediction error was split into is constituent parts. Valence refers to the sign of the prediction error (better or worse than expected), while surprise refers to the magnitude of the deviation between expectation and observations. We found evidence for signed prediction errors for both the correct and incorrect policy. Interestingly, we found an interactive effect between valence and surprise only for the incorrect policy (Fig A in S1 Supporting Information). These results indicate that valence is not separately processed for the correct policy, but distinctly modulates the processing of surprise for the incorrect policy. Given the novelty of the task, this intricate pattern is difficult to reconcile. Future studies should more closely investigate the temporal and spatial patterns of these effects to learn more about the underlying cognitive mechanisms.

In sum, the overall pattern indicates that central model variables involved in adaptive learning and credit assignment are reflected in neural activity. Taken together, these results allow for two conclusions. First, they validate the first-level policies implemented in our model architecture and suggest that reward prediction errors for both correct and incorrect policies are in fact concurrently calculated in the brain. Moreover, our findings suggest that the evidence signal postulated by our model is indeed computed in the human brain, which validates the hypothesized second-level inference process for solving the structural credit assignment problem.

## Brain activity reflects feedback-locked evidence signals and response-locked arbitration weights for credit assignment

In the feedback-locked analysis the evidence signal was clearly reflected in the neural data, whereas the arbitration weight did not produce a significant cluster. Complementing the finding that arbitration weights but not evidence signals drive future choices, we sought to test the hypothesis that arbitration weights are realized at the response level. To reveal the response-

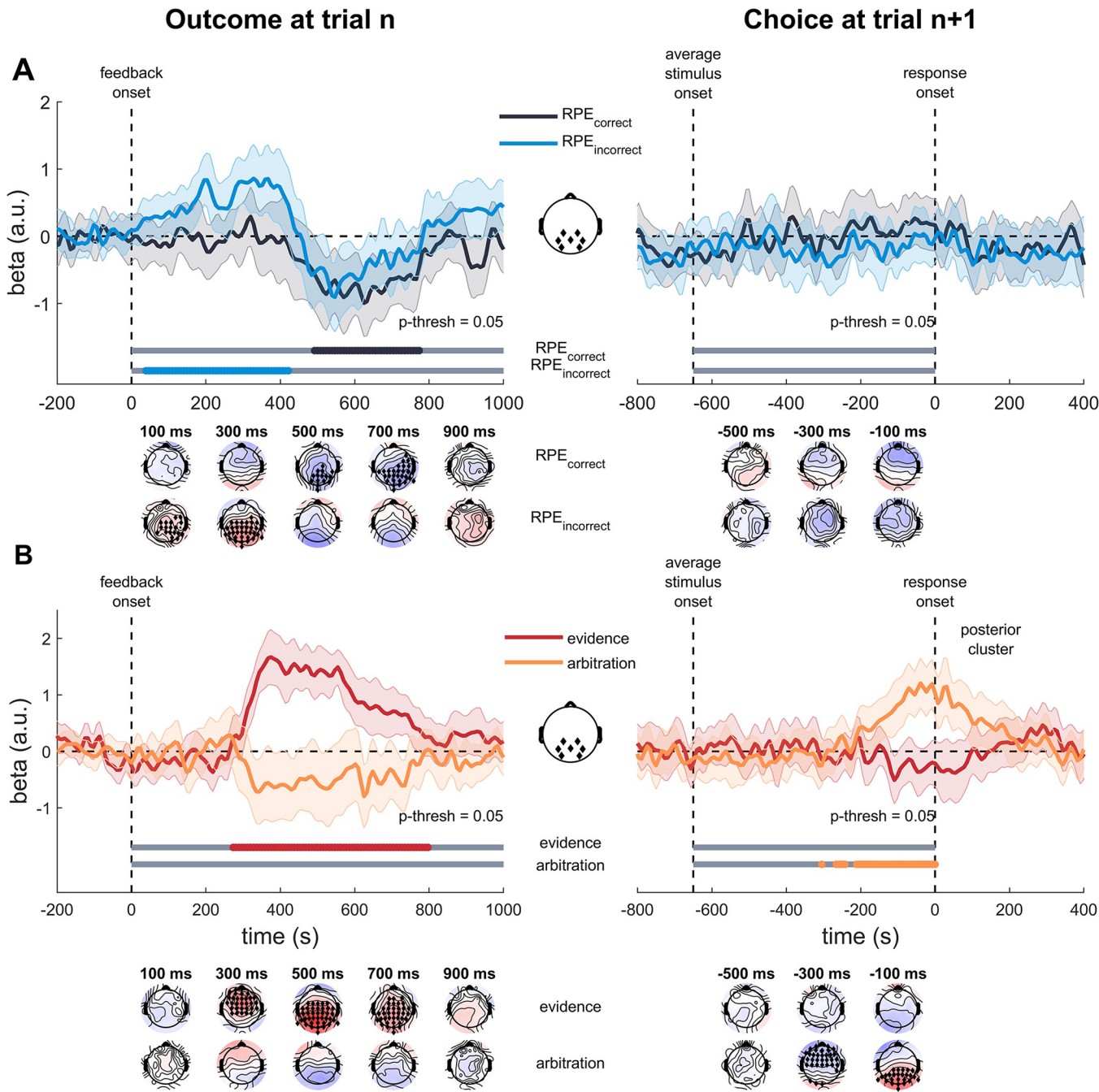

**Fig 5. Model-based regression analyses.** A. Mean regression (beta) values for prediction errors for the correct and incorrect policies, separately for the feedback-locked data (left panel) and response-locked data (right panel) at posterior electrode sites. B. Mean regression (beta) values for the evidence signal, calculated as the difference between correct and incorrect surprise and the arbitration weight, calculated as the inverse logit-transformed accumulated evidence signal. Gray bars below indicate the time windows which were considered for cluster-based permutation testing. Colored bars indicate time windows with significant positive and negative effects. The posterior cluster was defined by electrodes Pz, P3, P4, CP1, CP2, PO3, and PO4, as indicated in the central inlay. Topographies show the significant cluster for the correct policy and the incorrect policy. Black diamonds indicate significant clusters.

locked nature of the assignment weight, we replicated the previous regression approach, but predicted neural activity before response onset. We found that neither prediction errors for the different policies nor the evidence signal was reflected in the response-locked data.

Crucially, the arbitration weight produced significant clusters. The first cluster extended from approximately -414 to -140 ms at frontal locations whereas the second cluster extended from -315 to 0 ms at parieto-central locations. This finding suggests that arbitration weights are utilized only during the choice process, contributing to the weighting of the action values from the different policies.

## Discussion

In the present study, we introduced a novel model for structural credit assignment in the human brain. A central principle of this model is an arbitration mechanism which compares surprise signals from alternative representations of decision-outcome mappings and weights their respective policies so that action selection minimizes surprise. In a novel version of the bandit task, in which multiple decisions and multiple outcomes induce ambiguity about the causal structure, we tested human participants on their ability to infer the hidden structure of the task and interpreted the results in accordance with our computational model. Behavioral data from both implicit and explicit measures showed clear hallmarks of credit assignment. Crucially, our computational model with arbitration towards minimizing surprise outperformed alternative models without such a mechanism. Moreover, it revealed the dynamics of credit assignment and successive arbitration towards the correct mapping in the behavioral data. In line with the idea that behavioral arbitration reflects an incremental and gradual process, we found evidence of behavioral control of correct and incorrect representations in both model free and model-based analysis of the behavioral data. Finally, our electrophysiological results supported our predictions about independent, competing reinforcement learning policies and surprise-based inference by demonstrating that latent variables associated with these processes are reflected in neural activity on a trial-by-trial basis after feedback and before response onset.

The concept of surprise plays a major role in multiple lines of neuroscientific research and has greatly advanced our understanding of the human brain as an information processing system. As promoted within reinforcement learning frameworks, surprise is most commonly conceptualized as the prediction error and constitutes a teaching signal for the updating of values [e.g., 1,33,34]. Such a direct utilization for value learning necessitates information about valence that is carried only in signed prediction errors. But surprise is also able to inform indirect processes that support structure learning and decision-making more broadly [e.g., 35]. For example, surprise can signal that the brain's current model of the world is inaccurate. Interestingly, the necessary information can be signaled by unsigned predictions errors. In non-Bayesian approaches, surprise consequently drives the speed of learning via the dynamic scaling of the associability (i.e., learning rate) parameter [25,36, but also 37]. In Bayesian approaches, surprise modulates the precision of prior beliefs [38,39]. Such a precision-weighting mechanism has been shown to drive human learning and decision-making via dopaminergic modulation and is impaired in psychosis [40]. The mechanism of precision-weighting clearly resonates in different neurobiological theories. For example, the "mixture of experts" framework [41] proposes that multiple different systems cooperate and compete for behavioral control. As a central mechanism, the degree of reliability as carried in the surprise component of the prediction error is a necessary and sufficient component to allocate control weights over experts to optimize external payoffs and internal cost-benefit tradeoff. Within the predictive coding or free energy frameworks [42–44], surprise is treated as negative model evidence and under suitable assumptions free energy even is formally equivalent with confidence-weighted prediction errors [45]. Minimizing surprise is then used to arbitrate between competing representation and belief states of the world. Our computational model convincingly realized

crucial proposals: over time the impact of unreliable experts or representations on behavior is reduced and leads to the emergence of an adequate world model, task representation and behavioral adaptation.

Contrasting with the importance of surprise for structural inference, the exact nature of how surprise is encoded in the brain is still a matter of debate [46]. In the current study, the obtained clusters of surprise-related neural activity were evident over posterior and central electrode site. Both the first-level variables (prediction errors) as well as the second-level variable (evidence signals) coincide with activity known as the P3 component of the human ERP. Interestingly, this component has been repeatedly linked to the processing of surprise [47–51]. In a seminal study [51], the P3 following an oddball stimulus was shown to track surprise as formalized by information theory and could therefore be attributed to estimations of the probabilities of upcoming events by an ideal observer. Recent evidence further suggested that the amplitude of the P3 is dependent on the source of surprise, as indicated by an intricate neural pattern for different learning contexts [52]. Using a predictive inference task, this study demonstrated that the brain keeps track of the uncertainty associated with an outcome to adjust learning adaptively. Posterior activity reflected in the P3 was shown to predict enhanced learning if surprise minimization via behavioral selection was possible but predicted reduced learning if surprise minimization was impossible. By showing that similar posterior activity might reflect an evidence signal distinguishing between different task representations in our multiple-bandits task, our findings add to this literature on the connection between posterior activity and surprise-based inference about the task structure [53].

Our EEG results also demonstrate the existence of multiple, often concurrent task representations as reflected by different (signed) prediction errors. Support for the existence of multiple task representations in the brain comes from several recent lines of research. Studies on hierarchical reinforcement learning could show that complementary to reward prediction errors, pseudo-prediction errors reflect the achievement of subgoals. This finding suggests that complex tasks can be divided into multiple subtask representations [54,55]. Furthermore, studies investigating the distinction between model-free and model-based reinforcement learning indicate that the brain reflects prediction errors from both task representations [23,29]. Most relevant for the current study is the finding that the neural signatures of prediction errors are dependent on the representation of the task context [56]. Using an instrumental learning task in which participant could either take or forfeit a gamble, Fischer & Ullsperger show that late parietal covariations of prediction errors are opposed in sign for the different actions. While this finding does not necessitate the existence of multiple task representations, it clearly suggests how multiple, opposing prediction errors can be calculated from the same task outcome in a counterfactual manner.

Phasic dopamine responses have long been assumed to encode a scalar prediction error value as calculated by a single temporal difference learning agent [57,58]. However, the growing wealth of studies showing the existence of multiple simultaneous prediction error calls for a reinterpretation of the role of dopamine in signaling prediction errors [59]. In line with this proposal, recently published findings suggest that dopamine function might actually be better understood as retrospective causal inference (i.e., credit assignment), contesting the prospective temporal difference reinforcement learning framework [60]. While this work calls the prevailing reward prediction error hypothesis of mesolimbic dopamine into question, its proposed model is highly consistent with and reliant on the notion of surprise to solve the (temporal) credit assignment problem. As such, surprise-minimization can be interpreted as an additional mechanism, specialized to solve structural credit assignment.

Credit assignment and inference in the human brain has recently been investigated in other paradigms than the present one. Most of these studies employed a version of the simple bandit

task with one action and one outcome [13,61–63]. In these instantiations of the task, the trial structure inherently carries information about the causal structure of the environment with no hidden structure, thus lacking the need for representation learning. Noteworthily, two studies deviated from this approach. First, in a task resembling traditional free operant paradigms, participants were faced with decisions that were available only for a short amount of time [12]. As outcomes followed the action with a fixed time delay, the relevant outcome for a given action could sometimes occur only after the next decision was presented. Although participants were instructed on the contingency between action and outcomes, behavioral adaptation was found to reflect a mixture of contingent and non-contingent learning mechanisms. Using fMRI, it was demonstrated, that the orbitofrontal cortex has a guiding role in the arbitration between these mechanisms and balances them via suppression [12]. Second, in a latent social cause inference task, participants had to indicate which of two presented human agents was responsible for the outcome on each trial [64]. Using computational modeling and fMRI in this task with hidden states, the authors highlight separate processes involved in solving the causal inference problem: while the updating of association between cause and outcome takes place in the dorsomedial prefrontal cortex, causal inference is decoded in hippocampal activity. In line with our conclusion that multiple task representations compete for action selection to guide credit assignment, the authors of both studies emphasize that surprise and uncertainty about the causal structure is the main reason for the application of multiple learning mechanisms and processes.

Taken together, our multiple-bandits task allowed us to investigate structural credit assignment and representation learning in human participants. By applying a novel computational model which arbitrates action control towards the task representation that minimizes surprise, we were able to highlight basic principles that underlie credit assignment in the brain and reveal neural correlates of the involved core mechanisms.

## Methods

### Ethics statement

Ethical approval for this study was granted by the Ethics Committee of the Catholic University of Eichstätt-Ingolstadt. All participants provided written informed consent prior to the experiment.

### Experimental model and subject details

Experiment 1 (behavioral) included 48 participants (42 female) between 18 and 30 years of age ($M$ = 21.02, $SD$ = 2.82) and Experiment 2 (EEG) included 28 participants (21 female) between 18 and 28 years of age ($M$ = 20.54, $SD$ = 2.33) 2. All participants had normal or corrected-to-normal vision, were recruited at the Catholic University of Eichstätt-Ingolstadt and received course credit for participation and a performance-dependent bonus (Experiment 1: $AM$ = 0.55 €, $SD$ = 0.48 €; Experiment 2: $AM$ = 0.87 €, $SD$ = 0.54 €).

### Method details

**Stimuli.** Stimuli were comprised of 32 black-and-white images of everyday objects obtained from the online database of the International Picture Naming Project [65]. Pictures were converted into 300x300 pixel images with a side length of 4˚ visual angle at a viewing distance of 70 cm. For each block, four pictures were randomly sampled from the stimulus array (without replacement). These four draws were then divided into two subsets with two pictures each. These subsets were randomly assigned to constitute the first and second decision

stimulus pair for the upcoming tasks. For each stimulus pair, its respective pictures were shown to the left and right of a centrally presented fixation cross at a distance of 1.6˚. The left or right position for each stimulus in a pair was randomly determined for each trial. The outcome stimuli were colored two-digit numbers. Outcome colors were yellow (RGB = 170,170,0) and blue (155,155,255), controlled for isoluminance. In Experiment 1, outcome stimuli (0.6˚ height, 0.4–1˚ width) were presented simultaneously in two non-overlapping locations on the screen. On each trial, the two outcome locations were randomly drawn without replacement from 16 candidate locations, equally spaced across the screen in a 4x4 matrix (6.8˚ x 6.8˚). During outcome presentation, the fixation cross was presented centrally. In Experiment 2, outcome stimuli (2.2˚ height, 0.8–3.3˚ width) were presented one after another at the center of the screen in random order. All stimuli were presented on a black background.

**Multiple-bandits task.** Participants in both experiments performed the multiple-bandits task, a modified version of the two-armed bandit task (Fig 1B). The goal in this task is to maximize outcomes by repeatedly making decisions between different stimuli or actions. Critically, on each trial in the multiple-bandits task (Fig 1C and 1D), participants made not one but two separate decisions and hence received two separate outcomes. For each decision, participants were presented with a stimulus pair and asked to take an action for one of its stimuli. The action was taken by pressing one of two keys on an English standard keyboard (E with the left index finger for choosing the left stimulus, I with the right index finger for choosing the right stimulus). There was no time limit for responding. The interval between the action of the first decision and the stimulus of the second decision varied randomly between 500 and 1000 ms. A fixation cross was presented during both decisions. In Experiment 1 (Fig 1C), the interval between the second decision and subsequent outcome presentation again varied randomly between 500 and 1000 ms. In Experiment 2 (Fig 1D), this interval was constant at 1000 ms. As a trial consisted of two decisions, two outcomes were presented. The outcome that could be delivered from a specific decision ranged between 1 and 99 points with distinct outcome values $P$ and $(100 - P)$, dependent on the action. $P$ varied throughout a block as a random walk process. For the first trial, P was randomly drawn uniformly from the interval [1,99]. On each subsequent trial, $P$ was updated by adding a value sampled from a Gaussian distribution with $M = 0$ and $SD = 15$ (Fig 1B). This means that outcomes remain stable (i.e., above or below the win/loss threshold of 50 points) for 5.7 trials on average. Because a trial consisted of two separate decisions, the same random walk procedure was applied for the first and the second decision, and both probability distributions were independent. Outcomes were color-coded in blue and yellow. For example, the possible outcomes of the first decision $P$ and $(100 - P)$ were associated with a blue outcome stimulus whereas the possible outcomes of the second decision $Q$ and $(100 - Q)$ were associated with a yellow outcome stimulus. For Experiment 1, the two outcomes were presented simultaneously for 2000 ms at random, non-overlapping locations on the screen. To be able to discern the neural correlates for feedback processing, the two outcomes in Experiment 2 were presented consecutively (rather than simultaneously) for each 1000 ms in a random order with 1000 ms interstimulus interval. At the end of each trial, a fixation cross displayed for a random time between 500–1000 ms led to a screen that asked participants to press the space bar to continue to the next trial.

**Transfer task.** After each block of the multiple-bandits task, participants worked through a transfer task in which we directly assessed the participant's knowledge about the mapping between decision and outcome color of the previous multiple-bandits task. To this means, we compiled new stimulus pairs by recombining the stimuli used in the two decisions. The new stimulus pairs always contained one stimulus from each decision's stimulus pairs, resulting in four possible combinations of stimuli. On each trial, a random new combination was presented centrally, using the same method as for stimulus presentation in the multiple-bandits

task. Again, an action for a stimulus had to be taken by pressing one of two keys on an English standard keyboard (E with the left index finger for choosing the left stimulus, I with the right index finger for choosing the right stimulus). An action was required within 3500 ms after stimulus onset. If no action was taken within this time limit, the trial was counted as a miss and the stimulus disappeared. Participants were informed that misses resulted in a loss of 3 cents. If an action was taken within the time limit, the stimulus also disappeared. In Experiment 1, the disappearance of the new stimulus was followed by the presentation of a fixation cross for a random time of 500–1000 ms. For Experiment 2, the fixation screen stayed on screen for a fixed duration of 1000 ms. Afterwards, the next trial of the transfer task started. We instructed participants to choose the stimulus that had previously been associated with a specific target color (blue or yellow). For each block, the target color was randomly chosen.

**Experimental structure.** Both experiments were identically structured, and were implemented using Presentation software (Neurobehavioral Systems, Albany, CA). Before starting the experiment, participants received written instructions on both the multiple-bandits task and the transfer task. Additionally, each task instruction was complemented by a training session consisting of 10 trials for the multiple-bandits task and 12 trials for the transfer task. Note that during the training of the transfer task, participants received feedback about their actions. The multiple-bandits task was administered in blocks of 100 trials. The position of decisions (first or second) and the connection between decision and color (yellow or blue) was fixed within a block but randomized between blocks. The transfer task was administered in blocks of 20 trials, in which no feedback about the validity of the participants' decision was presented. Participants performed 3 blocks in a session, whereas each block consisted of one block of the multiple-bandit task and one block of the transfer task. In the instruction and training, participants were familiarized with the two key features of the random walk procedure: anti-correlation and continuity. Anti-correlation was introduced by the statement that within one decision there are always exactly 100 points distributed between the stimuli. Continuity was introduced by the statement that on each trial only a fraction of the coins is re-assigned between stimuli within each decision. Participants were also provided with a tabular example of such an anti-correlated and continuous random walk. Moreover, participants were informed about the mapping between decisions and feedback colors. Crucially, participants were never instructed on the correct mapping between decisions and outcome color but had to infer the decision-outcome mapping themselves in each block.

**Electrophysiological recordings.** Participants were seated comfortably in a dimly lit, sound-attenuated and electrically shielded cabin. The electroencephalogram (EEG) was recorded using a BIOSEMI Active-Two system (BioSemi, Amsterdam, The Netherlands) with 64 Ag-AgCl electrodes from channels Fp1, AF7, AF3, F1, F3, F5, F7, FT7, FC5, FC3, FC1, C1, C3, C5, T7, TP7, CP5, CP3, CP1, P1, P3, P5, P7, P9, PO7, PO3, O1, Iz, Oz, POz, Pz, CPz, Fpz, Fp2, AF8, AF4, AFz, Fz, F2, F4, F6, F8, FT8, FC6, FC4, FC2, FCz, Cz, C2, C4, C6, T8, TP8, CP6, CP4, CP2, P2, P4, P6, P8, P10, PO8, PO4, and O2, as well as the left and right mastoid. All electrodes were placed according to the extended International 10–20 EEG system. The horizontal and vertical EOG was monitored by means of four electrodes, placed above and below the right eye and the outer canthi of both eyes. Sampling rate was 512 Hz.

EEG data were preprocessed and analyzed using custom-made routines in MatLab as well as EEGLAB 13.5.4b [66], an open source toolbox for EEG data analysis (EEGLAB toolbox for single-trial EEG data analysis, Swartz Center for Computational Neurosciences, La Jolla, CA; http://www.sccn.ucsd.edu/eeglab). EEG data were offline re-referenced to averaged mastoids, band-pass filtered to exclude frequencies outside the range 0.5–35 Hz and divided into epochs separately for choice and feedback. Response-locked epochs reached from 1000 ms before to 500 ms after response onset, separately for each of the two decision stages in each trial. Baseline

activity was removed by subtracting the average voltage from 200–0 ms before stimulus onset. Feedback-locked epochs reached from 500 ms before to 1000 ms after outcome onset, separately for each of the two outcomes in each trial. Baseline activity was removed by subtracting the average voltage from 200–0 ms before outcome onset. Bad channels were interpolated using spherical spline interpolation if they met the joint probability criterion (threshold 5) as well as the kurtosis criterion (threshold 5) using EEGLAB's channel rejection routine. Furthermore, epochs were excluded for which neural activity in a channel exceeded a ±300 μV threshold from the epoch mean. This criterion was not applied to those channels that are typically contaminated by blinks (Fp1, Fpz, Fp2, AF7, and AF8). An infomax-based independent component analysis (ICA, ref. [67]) was conducted to derive independent components. After visual inspection of these components, those representing eye blinks and muscular artifacts were identified and removed from the data.

**Quantification and statistical analysis.**  All statistical analyses and computational modeling results in the results and figure legends were realized using custom-written routines in MatLab v8.6 (The Mathworks Inc., Natick, MA).

**Behavioral data analysis.**  We performed identical analyses of behavioral data for Experiment 1 and Experiment 2. To investigate whether participants correctly inferred the correct decision-outcome mapping, the resulting choice behavior in the multiple-bandit task was analyzed using logistic regression with a mixed-effect model. Central goal of the regression was to predict upcoming behavior based on the previous outcomes. As in standard bandit tasks, choice behavior in the multiple-bandits task is characterized by win-stay/lose-shift pattern, i.e., agents tend to repeat choices after wins but switch choices after losses. We considered both decisions in this analysis, that is, two decisions per trial were included. Each action in the multiple-bandits task was classified as either a *stay* or a *switch* (coded as 1 or 0), depending on the previous action for the same decision. For the regressors, each outcome (ranging between 1 and 99) was first defined as a win or a loss (coded as 1 or -1), depending on whether its value was larger or smaller than 50, respectively. As a first regressor, we included the relevant outcome on the previous trial, that is, the outcome that was assigned to the decision under consideration. Depending on this outcome, a given decision of a trial was categorized as being preceded by either a *relevant win* or a *relevant loss* (coded as 1 and -1). As a second regressor, we included the irrelevant outcome on the previous trial, that is, the outcome that was assigned to the other decision. Depending on this outcome, a given decision of a trial was categorized as being preceded by either an *irrelevant win* or an *irrelevant loss* (coded as 1 and -1). Please note that this procedure implies that each of the two outcomes of trial was used to predict each of the two actions in the upcoming trial, once as the relevant outcome and once as the irrelevant outcome. For each participant, a logistic regression was calculated,

$$\text{Stay} \sim b_0 + b_1 * \text{relevant} + b_2 * \text{irrelevant} + b_3 * \text{interaction.} \qquad \text{(Eq 1)}$$

For Experiment 2, one participant (VP 20) was excluded from the analysis, as the regression model failed to converge on a stable solution. Across participants, the resulting regression weights were tested against zero, using one-sample t-tests.

While the logistic regression analysis can show whether relevant and irrelevant outcomes influenced choice behavior, they cannot tell us whether the relevant outcome has a stronger impact than the irrelevant outcome. We therefore carried out a planned contrast analysis to directly quantify implicit credit assignment in the multiple-bandits task. We reasoned that unitary outcomes (relevant win/irrelevant win and relevant loss/irrelevant loss trials) do not require credit assignment to optimize action selection. Following the previously introduced idea of win-stay/lose-shift, double wins and double losses should always lead to the same stay/

shift behavior irrespective of whether credit assignment was successful or not. Therefore, we focused on the effects of mixed outcome, that is, trials on which the relevant and irrelevant outcome was different. For each participant, we aggregated stay probabilities for the two possible mixed outcome pairs (relevant win/irrelevant loss and relevant loss/irrelevant win trials) and compared them using the non-parametric Wilcoxon signed-rank test. We hypothesized that successful credit assignment is reflected in higher stay probabilities for the relevant win/irrelevant loss condition as compared to the relevant loss/irrelevant win condition.

In addition to these main analyses, we also tested performance of both the multiple-bandits task and the transfer task against chance level using the Wilcoxon signed-rank test. Performance in the multiple-bandits task was defined as the proportion of wins (i.e., choices that led to outcomes higher than 50) following the same dichotomization procedure for outcomes as in the logistic regression analyses. Performance in the transfer task was defined as the proportion of choices in line with the transfer instruction and taken as a measure of explicit credit assignment which complements the measure of implicit credit assignment from the multiple-bandits task. Finally, we analyzed correlations between the different measures of credit assignment and performance using Pearson correlations.

**Computational models.** To explore and verify the idea of surprise minimization for solving the credit assignment problem in our task, we introduced a novel computational model. The main challenge for any model solving the credit assignment problem in the multiple-bandits task is that, besides the tracking of changing outcome values (i.e., value learning), the causal structure of the task must be inferred (i.e., representation learning). At its core, our proposed model is a simple extension of traditional approaches to modeling two-armed bandit tasks in a reinforcement learning framework. However, due to the hidden nature of the decision-outcome mapping, we equipped our model with multiple independent reinforcement learning policies and implemented an inference mechanism, which infers the plausible representation of the task and arbitrates control for action selection towards the policy reflecting this representation.

On the first level, two reinforcement learning policies concurrently learn the values of actions, whereas each policy corresponded to one of the two possible mappings between decisions and outcomes in the task (Fig 1A): (1) The correct mapping links the decision $d_1$ to outcome $o_1$ and the decision $d_2$ to outcome $o_2$. (2) The incorrect mapping links decision $d_1$ to outcome $o_2$ and the decision $d_2$ to outcome $o_1$. Both the correct and the incorrect mapping are realized in an independent first-level policy, which captures the (hypothesized) causal structure and updates its values $V$ for the action $a$ taken at decision $d$. On each trial $t$, each policy calculates the prediction error $\delta$ for each decision using temporal difference learning,

$$\delta_{\text{correct}}(d_1, t) = r(o_1, t) - V_{\text{correct}}(d_1, a, t), \tag{Eq 2}$$

$$\delta_{\text{incorrect}}(d_1, t) = r(o_2, t) - V_{\text{incorrect}}(d_1, a, t), \tag{Eq 3}$$

where $r$ denotes the reward (i.e., outcome values) and $o$ denotes the outcome color. The equivalent equation is applied for decision $d_2$. The expected action values within each policy are then updated for the next trial as

$$V_{\text{correct}}(d_i, a, t + 1) = V_{\text{correct}}(d_i, a, t) + \alpha * \delta_{\text{correct}}(d_i, t), \text{ for } i \in [1, 2], \tag{Eq 4}$$

with $\boldsymbol{\alpha}$ reflecting the learning rate across policies. The equation is applied for both decisions $d_i$ ($i = 1, 2$), and an equivalent equation is applied for $V_{\text{incorrect}}$ for the incorrect policy. Additionally, we implemented counterfactual updating for the unchosen options for each policy. Counterfactual updating means that the scaled prediction errors were subtracted from the unchosen

option's value. This choice was motivated by the instructed anti-correlated nature of the random walks within each decision-outcome pair. Crucially, model selection clearly confirmed the mechanisms against instantiations with a learning rate-based decay $(1 - \alpha)$ or a separate learning decay parameter [e.g., 68,69] (see Table A in S1 Supporting Information), which represent alternative mechanisms to weaken action values of unchosen options.

While the computations for the two first-level policies are carried out independently and in parallel, the respective estimates about prediction errors are passed along to the second stage in form of a surprise signal for each decision that follows the idea of surprise [23,25] so that

$$\text{surprise}_{\text{correct}}(d_i, t) = |\delta_{\text{correct}}(d_i, t)|, \text{ for } i \in [1, 2]. \tag{Eq 5}$$

The equivalent equation is also applied for the incorrect policy. On the second level, an inference principle is implemented that utilizes the arriving surprise signals from each policy. The goal of inference is to base decision making on a policy that minimizes surprise for actions and it does so by successively arbitrating control towards the policy associated with the smallest surprise. Following the notation that surprise is negative model evidence, the most plausible policy for the task at hand is the policy with the least surprising (and thus most precise) estimations of action values and consequently the smallest absolute prediction errors. Critically, the second-level inference mechanism generates an evidence signal $\Delta$ for each outcome by comparing surprise signals between the candidate policies as follows

$$\Delta(o1, t) = \text{surprise}_{\text{incorrect}}(d2, t) - \text{surprise}_{\text{correct}}(d1, t) \tag{Eq 6}$$

$$\Delta(o2, t) = \text{surprise}_{\text{incorrect}}(d1, t) - \text{surprise}_{\text{correct}}(d2, t) \tag{Eq 7}$$

Please note that for this computation, surprise signals for different decisions have to be compared in order to account for the evidence for each outcome. Crucially, this calculation is most important for the model-based analysis of the electrophysiological data, as it makes a testable prediction for the validity of the surprise minimization model. After comparison of the surprise signals between policies, this evidence signal is integrated into the arbitration weight

$$\omega(t + 1) = \omega(t) + \varepsilon * \sum_i \Delta(o_i, t), \quad for\ i \in [1, 2], \tag{Eq 8}$$

with $\varepsilon$ being the assignment rate used for updating the accumulated evidence. The arbitration weight is initialized as 0. Model selection clearly confirmed the mechanisms against instantiations with a free starting point different from 0 (see Table A in S1 Supporting Information). Translating the policy-specific surprise signals into an arbitration weight allows action selection on each trial to be based on the weighted net values of actions,

$$V_{\text{net}}(d_i, a, t) = \text{logit}^{-1}(\omega(t)) * V_{\text{correct}}(d_i, a, t) + (1 - \text{logit}^{-1}(\omega(t))) * V_{\text{incorrect}}(d_i, a, t), \text{for } i \in [1, 2]. \tag{Eq 9}$$

For the arbitration weight to span between 0 and 1, the inverse logit transformation was applied. Choice probabilities for each decision are calculated according to the softmax rule:

$$P\left(d_i, a_j, t\right) = \frac{\exp(\beta * V_{\text{net}}(d_i, a_j, t) + \rho * \text{rep}(d_i, a_j))}{\sum_{j=1}^{2} \exp(\beta * V_{\text{net}}(d_i, a_j, t) + \rho * \text{rep}(d_i, a_j))}, \text{for } i \in [1, 2], \tag{Eq 10}$$

with $\beta$ resembling the inverse temperature parameter and $\rho$ resembling the perseveration parameter. The indicator function $rep(d_i, a)$ is defined as 1 if $a$ is the action that was chosen for the same decision in the previous trial and 0 otherwise. At the beginning of a block, action values for each policy, decision and action and the arbitration weight were initialized as 0.

Moreover, outcome values, originally spanning between 1 and 99, were centered to span between -1 and 1 (r = (r/50)-1).

Besides the surprise minimization model, we also considered four alternative model for subsequent competitive model fitting. The first three models (correct policy model, incorrect policy model and random policy model) are nested versions of the surprise minimization model. All models follow the exact same procedure of action value updating as described above. The central difference between these models is that (1) the assignment rate is preset to 0 and (2) the arbitration weight is fixed for each instantiation of the model. For the correct policy model, the arbitration weight is fixed at 0.9, so that choice behavior always follows the correct mapping between decision and outcome. For the incorrect policy model, the arbitration weight is fixed at 0.1, so that choice behavior always follows the incorrect decision-outcome mapping. For the random policy model, the arbitration weight is fixed at 0.5, so that choice behavior is always equally balanced between the correct and the incorrect mapping.

The fourth model, called the joint action model [70], is a standard temporal difference model without a second-level inference mechanism. Crucially, the model is defined on the space of joint actions. At the end of each trial of the multiple-bandits task a scalar prediction error is calculated as

$$\delta(t) = (r(o1, t) + r(o2, t)) - V(j, t) \tag{Eq 11}$$

with *j* representing the combination of the two actions in a trial. More specifically, *j* can take four possible values, corresponding to the combination of the (two) possible first decision actions and the (two) possible second decision actions. Updating of actions values for each possible joint action and action selection via softmax follows the identical procedure as introduced in the surprise minimization model.

**Monte-Carlo simulation and model comparison.** For the simulation, we set up 1000 runs in the same instantiation of the multiple-bandits task as in the previous behavioral experiment with 3 blocks and 100 trials per each block. Parameters for each run were randomly drawn from empirical prior distributions [71]. The learning rate was bound in [0; 1] and the respective prior was $\alpha \sim$ beta(1.2, 1.2). The inverse temperature was bound in [0; 20] and the prior was $\boldsymbol{\beta} \sim$ gamma(2, 1). The perseveration parameter was bound in [-5; 5] and the prior was $\rho \sim$ gauss(0, 1). The assignment rate $\varepsilon$ was bound in [0; 1] and the respective prior was $\varepsilon \sim$ beta(1.2, 1.2). To separately explore the functioning of the first-level policies and the second-level inference mechanism, we simulated the correct policy model, the incorrect policy, and the full surprise minimization model. Crucially, to prevent spurious differences, the simulations of the choice patterns for all three models were based on the same model parameter values and seeds of the Gaussian random walk.

For parameter fitting, we implemented all four candidate models separately and fitted them to the observed behavioral data using the mfit toolbox [71]. Parameters were fitted for each participant separately using the non-uniform priors and specific bounds, as specified for the Monte-Carlo simulation. Three measures were used for model comparison: the Bayesian information criterion (*BIC*), the Akaike information criterion (*AIC*) and the exceedance probability (*xp*). The latter represents the belief about the posterior probability of a parameter and is highest for the model providing the best fit to the data [26].

After model comparison, the variables from the best fitting model ($V_{\mathbf{correct}}$, $V_{\mathbf{incorrect}}$, $\Delta$, $\mathbf{logit^{-1}(\omega)}$) were used to predict the behavioral choice on the next trial of the multiple-bandits task. The linear equation for the mixed random-effects model took the following form

$$\text{choice} \sim b_0 + b_1 * D_{\text{correct}} + b_2 * D_{\text{incorrect}} \tag{Eq 12}$$

$$+b_3*\Delta + b_4*\text{logit}^{-1}(\omega)$$

$$+b_5*D_{\text{correct}}*\Delta + b_6*D_{\text{incorrect}}*\Delta$$

$$+b_7*D_{\text{correct}}*\text{logit}^{-1}(\omega) + b_8*D_{\text{incorrect}}*\text{logit}^{-1}(\omega)$$

$$+\text{error},$$

with D reflecting the decision value, calculated as the difference between action values (V(a1) minus V(a2)), reflecting the direction and difficulty of the decision. The random effects structure included random intercepts and slopes for all variables and interactions per participant.

Finally, the parameters of the best fitting model ($\alpha, \beta, \rho, \varepsilon$) were used to predict the behavioral measures of performance in the multiple-bandits task and the transfer task, as well as the measure of implicit credit assignment in separate linear regressions. The linear equation for each measure took the following form

$$\text{measure} \sim b_0 + b_1*\alpha + b_2*\beta + b_3*\rho + b_4*\varepsilon + \text{error}. \tag{Eq 13}$$

**Model-based analysis of electrophysiological data.** For the model-based single-trial analysis, we constructed two general linear models (GLM) in which the central latent variables from the surprise minimization model were used to predict downsampled (125 Hz), single-trial feedback-locked and response-locked EEG activity at each electrode and time point, separately for each participant. Latent variables were simulated by feeding the estimated model parameters back into the surprise minimization model.

For the first model, regressors included the prediction errors related to the correct and the incorrect policy, evidence signals and arbitration weights and the response variable is the feedback-locked activity. The linear equation took the following form for each participant,

$$\text{feedbackEEG} \sim b_0 + b_1*PE_{\text{correct}} + b_2*PE_{\text{incorrect}} + b_3*\Delta + b_4*\text{logit}^{-1}(\omega) + \text{error}. \tag{Eq 14}$$

For the second model, we used the same regression as above (Eq 14) to predict response-locked activity. The linear equation took the following form for each participant,

$$\text{responseEEG} \sim b_0 + b_1*PE_{\text{correct}} + b_2*PE_{\text{incorrect}} + b_3*\Delta + b_4*\text{logit}^{-1}(\omega) + \text{error}. \tag{Eq 15}$$

Please note that the evidence signal ($\Delta$) at feedback and response is calculated differently. While feedback-locked evidence follows the Eqs 5 and 6, response-locked evidence is calculated as a recombination of the four surprise signals. Moreover, due to the structure of the multiple-bandits task, each trial contributed two independent, event-locked signals, one for each feedback and response respectively. However, due to the architecture of the latent variables in the surprise minimization model, each event-locked signal can be assigned an individual latent variable.

Prior to regression, all predictors were z-scored. The resulting individual regression weights from each variable were standardized by their respective standard deviation to ensure comparability between participants and to penalize multicollinearity between predictors. Only after this normalization procedure, the individual beta weights were separately tested against zero via two-tailed cluster-based permutation tests implemented in the Mass univariate ERP Toolbox [72], conducted across sampling points and electrodes. We applied corrections for a family-wise alpha level of 0.05 and identified clusters for all sampling points at which the uncorrected p value fell below 0.05. To obtain sufficient test distributions, we implemented $10^4$

permutations. For feedback-locked analyses, all sampling points between 0 and 1000 ms after outcome presentation and all electrodes were considered for permutation testing. For response-locked analyses, all sampling points between -500 and 0 ms prior to response and all electrodes were considered for permutation testing. For clarity, we will report cluster-based permutation results following current guidelines [73]. That means, any statement about temporal and spatial extend are merely descriptive.

## Supporting information

**S1 Supporting Information. File containing supporting figures and tables. Table A: Model comparison for the different model variants with surprise minimization.** In a separate analysis, we identified the model with the surprise-minimization that best explains the behavioral data. We considered three different mechanisms for updating unchosen options: (1) Learning-rate based decay, (2) free-parameter decay and (3) counterfactual updating. Counterfactual updating is supported by the anti-correlated nature of the random walk procedure for each decision, as well as the model selection procedure (XP = 0.71). **Fig A: Model-based regression analysis for reward prediction errors split into valence and surprise components.** In a separate model-based EEG analysis we split the (signed) prediction errors in its constituent parts–valence and surprise–for both the correct and incorrect policy.
(PDF)

## Acknowledgments

We thank our thesis students and interns who helped with piloting and data collection.

## Author Contributions

**Conceptualization:** Benjamin Ernst, Marco Steinhauser.

**Data curation:** Franz Wurm.

**Formal analysis:** Franz Wurm.

**Investigation:** Franz Wurm.

**Methodology:** Franz Wurm, Benjamin Ernst, Marco Steinhauser.

**Project administration:** Franz Wurm, Marco Steinhauser.

**Resources:** Franz Wurm, Marco Steinhauser.

**Software:** Franz Wurm.

**Supervision:** Marco Steinhauser.

**Validation:** Franz Wurm.

**Visualization:** Franz Wurm.

**Writing – original draft:** Franz Wurm.

**Writing – review & editing:** Franz Wurm, Benjamin Ernst, Marco Steinhauser.

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
