## [Decision Letter · Decision Letter 0]

13 Nov 2023

Dear Dr Wurm,

Thank you very much for submitting your manuscript "Surprise-minimization as a solution to the structural credit assignment problem" for consideration at PLOS Computational Biology. Please accept my sincere apologies for the delay in processing the manuscript. We have unfortunately encountered difficulties in securing the reviewers until recently.

As with all papers reviewed by the journal, your manuscript was reviewed by members of the editorial board and by two independent reviewers. In light of the reviews (below this email), we would like to invite the resubmission of a significantly-revised version that takes into account the reviewers' comments.

We cannot make any decision about publication until we have seen the revised manuscript and your response to the reviewers' comments. Your revised manuscript is also likely to be sent to reviewers for further evaluation.

Sincerely,

Tianming Yang

Academic Editor

PLOS Computational Biology

Thomas Serre

Section Editor

PLOS Computational Biology

Reviewer's Responses to Questions

**Comments to the Authors:**

Reviewer #1: The review is uploaded as an attachment.

Reviewer #2: Wurm et al. studied neurocomputational mechanism of discovering decision-outcome mapping when it is hidden and needs to be discovered. For this purpose, the authors designed a task in which participants made two separate two-armed bandit choices and received separate outcomes, while the link between each decision and outcome was unknown and had to be discovered. The authors showed that participants learnt the task. They then designed a computational model based on competing policies: two different policies competed for explaining the behaviour. The policy that led to the smaller surprised had more influence on behaviour. The influence of each policy was determined by comparing the surprise of each policy. Using EEG, the authors indicated that neural activity was correlated with surprise obtained from the two competing policies (though it was greater for the correct policy). They also showed neural correlates of the difference between the surprise signals obtained from the two policies.

The study is interesting, and the research question is very timely. The manuscript is in general well-written, however I had trouble working out some behavioural analyses because of the lack of clarity in the way that the analyses were conducted and in the representation of the data. More importantly, I believe the authors do not provide adequate evidence for the “surprise minimisation” account which they try to advance. In addition, some of the statistical procedure seem incorrect to me. Please see my comments below.

1) In their introduction and discussion and the title of the paper, the authors suggest that surprise-minimisation is a solution and mechanism for solving the task. Their evidence for this account is the neural correlates of surprise that the study showed using EEG. I understand why the authors think “surprise minimisation” can be used to explain their results, because of the free energy principle. But if the authors wish to do so, they need to provide evidence that “surprise minimisation” is what the brain and the participants were doing for solving this task. The authors reviewed some articles related to the free energy principle. The free energy principle is not falsifiable (suggested by others and recently by its advocates too, please see https://www.sciencedirect.com/science/article/pii/S037015732300203X). So, my question is, how does this study provide evidence for the statement that the surprise minimisation is the solution for solving this task?

I recommend the authors to focus on the computational role of surprise to solve the task. Obviously, it is up to the authors to speculate about the consistency of their results with free energy principle, but it should not prevent them from presenting a more detailed computational account of their data. Critically, the surprise computed for each model can be used as the evidence in favour of each policy (as the authors compute). This is not the first study that investigates the effect of surprise in learning and decision making. Some other studies have investigated how surprise can be used to guide behaviour. For example, Harsmaa et al. Moceluar Psychiatry 2020 have studied the effect of surprise. In addition, Mahmoodi et al. BioRxiv 2023 have studied latent cause inference using surprise as a measure of solving the latent cause inference task. I think the authors should take a similar approach and discuss the relevant literature like those mentioned above. However, if they want to insist on “surprise minimisation” as it is suggested by the free energy principle, they have to provide evidence for their account, a feat which I’m afraid has been evading the advocates of the theory and eventually led to concede it is not a falsifiable theory.

By investigating the computational role of surprise in the task, the authors can ask more detailed and informative questions. For example, the difference in surprise signal (separately for each choice) can be used on the next trial as the “evidence signal”-- the probability that one policy is better than the other -- to guide the choice. The authors should run this analysis. The existence of such a signal indicates that the surprise signal is used to compute the likelihood that each policy is right. It will offer a computational role for surprise, similar to recent studies (e.g., Mahmoodi et al. BioRxiv 2023).

2) I found it very difficult to work out what Figure 2A-B and 3A are supposed to communicate. In figure 2, we have win/loss on x-axis, and we also have grey/white colours, but is not clear what each colour is. In addition, x-axis label indicates it is relevant outcome. moreover, on panel 2B there is a label at the top right which reads (win/loss for grey and white colours). How’s this labelling different from win/loss on x-axis?

3) As far as I understood, there is an effect of irrelevant win on stay probability. Is it not a failure of credit assignment? It indicates that if we you receive a good outcome, even it is irrelevant, you are more likely to repeat your current policy. This is a confusion effect. How do the authors explain it?

4) There is a study which indicates that the permutation test cannot be used to make any statement about the onset and offset of an effect (https://onlinelibrary.wiley.com/doi/10.1111/psyp.13335). I strongly urge the authors to report whether their procedure is consistent with this study. I am baffled by the significance of the red curve in Figure 5B, right panel. It does not seem significant to me even without applying any correction. The significance line indicates significance even for areas which the error bar almost overlaps with the zero line. How is that possible? I think the authors made the unwarranted interpretations of the permutation test that the study that I referred to above has addressed. If this is the case, the authors should correct their procedure.

**Have the authors made all data and (if applicable) computational code underlying the findings in their manuscript fully available?**

Reviewer #1: Yes

Reviewer #2: Yes

PLOS authors have the option to publish the peer review history of their article (what does this mean?). If published, this will include your full peer review and any attached files.

Reviewer #1: **Yes: **Jae Hyung Woo

Reviewer #2: No
---

## [Decision Letter · Decision Letter 1]

7 May 2024

Dear Dr Wurm,

Thank you very much for submitting your manuscript "Surprise-minimization as a solution to the structural credit assignment problem" for consideration at PLOS Computational Biology. As with all papers reviewed by the journal, your manuscript was reviewed by members of the editorial board and by several independent reviewers. The reviewers appreciated the attention to an important topic. Based on the reviews, we are likely to accept this manuscript for publication, providing that you modify the manuscript according to the review recommendations.

Sincerely,

Tianming Yang

Academic Editor

PLOS Computational Biology

Thomas Serre

Section Editor

PLOS Computational Biology

Reviewer's Responses to Questions

**Comments to the Authors:**

Reviewer #1: All of my comments were addressed adequately, and I appreciate the authors' efforts on re-running the analyses and fitting new models. Overall I think the paper has been strengthened. Below are just additional minor comments and follow-ups to the authors' response.

Comment 1: Thank you for checking this analysis. I agree that the new figure adds marginal value and better be excluded from the manuscript. Although, I find it very interesting that the psychometric curve for irrelevant mapping (red) seems to have overall stay bias (as the middle point at value of 50 seems to have over chance level of staying). Just wondering if the authors have insight to this effect?

For the average crossing rate (17.6 flips per 100 trials, stable for 5.7 trials) -- I think this detail is worth reporting and can be mentioned in the Methods for the task details.

Comment 2: Thank you for testing this additional models. I see how the additional starting bias is not helping with the fit, although decreasing the -log-likelihood. Table S1 is clear.

Figure 3E is also nice. A very minor styling preference: totally up to your choice, but since the third panel already says "Surprise" the y-label could be written as |δ|, consistent with the top panel. Y-label for Evidence generation could be Δ as well.

Also in the figure caption, it feels a little awkward that the order of explanation is from bottom to top panels, instead of top to bottom. Also there is a typo: "middle pane."

Comment 3: Thank you for testing this hypothesis. The results are interesting on their own, as they suggest that the brain is encoding the signal robustly. The new EEG analysis also shows nice separation between feedback and response periods. I refer additional comments for it to Reviewer 2.

Other additional comments:

- Typo in Eq. 4: dd_i should be d_i?

- Just caught this but in Eq. 9, the arbitration weight for the incorrect option should be (1-logit^-1(ω)) instead of logit^-1(1-ω), according to your implementation and the codes. Note that according to current formulation in Eq. 9, the two sigmoids (for correct & incorrect arbitration weights) are symmetric around ω of 0.5, not 0.

- Relatedly for Eq. 12, shouldn't the decision value for incorrect (D_incorrect) be multiplied with (1-logit^-1(ω)), instead of the current formulation which multiplies arbitration weight for "correct" decision value?

- Line #927: (Eq. 13) → (Eq. 14)

Reviewer #2: The authors have addressed most of my previous comments. However please see my remaining concern below.

I still find Figure 2A-B and 3A-C very confusing. For example, in Figure 2A we have win and loss on the x-axis. On the top right corner of Figure 2B, win and loss are color coded with dark and bright colours. On top of the color code it reads irrelevant outcome. I find these details very confusing. Can you please clarify these panels?

**Have the authors made all data and (if applicable) computational code underlying the findings in their manuscript fully available?**

Reviewer #1: Yes

Reviewer #2: Yes

PLOS authors have the option to publish the peer review history of their article (what does this mean?). If published, this will include your full peer review and any attached files.

Reviewer #1: **Yes: **Jae Hyung Woo

Reviewer #2: No

Figure Files:

Data Requirements:

Reproducibility:

References:

---

## [Editor Report · Decision Letter 2]

18 May 2024

Dear Dr Wurm,

We are pleased to inform you that your manuscript 'Surprise-minimization as a solution to the structural credit assignment problem' has been provisionally accepted for publication in PLOS Computational Biology.

Best regards,

Tianming Yang

Academic Editor

PLOS Computational Biology

Thomas Serre

Section Editor

PLOS Computational Biology

---

## [Editor Report · Acceptance letter]

22 May 2024

PCOMPBIOL-D-23-01284R2 

Surprise-minimization as a solution to the structural credit assignment problem

Dear Dr Wurm,

I am pleased to inform you that your manuscript has been formally accepted for publication in PLOS Computational Biology. Your manuscript is now with our production department and you will be notified of the publication date in due course.

With kind regards,

Zsofia Freund
